# BRCA1 and 2 Mutations and Efficacy of Pembrolizumab-Based Neoadjuvant Chemotherapy in Triple-Negative Breast Cancer: A Real-World Multicenter Analysis

**DOI:** 10.3390/jcm14248854

**Published:** 2025-12-14

**Authors:** Palma Fedele, Alessandro Rizzo, Matteo Landriscina, Stefania Luigia Stucci, Maria Morritti, Francesco Giuliani, Lucia Moraca, Giuseppe Cairo, Raffaele Ardito, Marianna Giampaglia, Domenico Bilancia, Assunta Melaccio, Antonella Terenzio, Antonio Gnoni, Antonella Licchetta, Federica Fumai, Laura Lanotte, Gennaro Gadaleta-Caldarola

**Affiliations:** 1Oncology Unit, Dario Camberlingo Hospital, Francavilla Fontana, 72021 Brindisi, Italy; coro.francavilla@asl.brindisi.it; 2S.S.D. C.O.r.O. Bed Management Presa in Carico, TDM, IRCCS Istituto Tumori “Giovanni Paolo II”, 70124 Bari, Italy; 3U.O. Medical Oncology and Biomolecular Therapy, Department of Medical and Surgical Sciences, University of Foggia, 71100 Foggia, Italy; matteo.landriscina@unifg.it (M.L.); antonella.terenzio@unifg.it (A.T.); 4Breast Care Unit, University Hospital Consortium Policlinico of Bari, 70124 Bari, Italy; stuccistefania@gmail.com; 5Oncology Unit, Fondazione “Casa Sollievo della Sofferenza”, IRCCS, San Giovanni Rotondo, 71013 Foggia, Italy; maria.morritti@gmail.com; 6Oncology Unit, San Paolo Hospital, 70123 Bari, Italy; francesco.giuliani@asl.bari.it (F.G.); assunta.melaccio@asl.bari.it (A.M.); 7Oncology Unit, Teresa Masselli Mascia Hospital, San Severo, 71013 Foggia, Italy; lucia.moraca@aslfg.it; 8Oncology Unit, Vito Fazzi Hospital, 73100 Lecce, Italy; giuseppecairo15@gmail.com; 9Day Hospital Oncologico IRCCS CROB, Rionero in Vulture, 85028 Potenza, Italy; raffaele.ardito@crob.it; 10Oncology Unit, San Carlo Hospital, 85100 Potenza, Italy; mariannagiampaglia@yahoo.it (M.G.); domenicobilancia@gmail.com (D.B.); 11Oncology Unit, Sacro Cuore di Gesù Hospital, Gallipoli, 73014 Lecce, Italy; antonio.gnoni@asl.le.it (A.G.); antonella.licchetta@asl.le.it (A.L.); 12Oncology Unit, Mons. Dimiccoli Hospital, 70051 Barletta, Italy; laura.lanotte@aslbat.it (L.L.); gennaro.gadaleta@aslbat.it (G.G.-C.)

**Keywords:** pCR, breast cancer, TNBC, neoadjuvant, pembrolizumab, triple-negative breast cancer

## Abstract

**Background:** Pembrolizumab has reshaped the neoadjuvant treatment landscape for triple-negative breast cancer (TNBC). However, the influence of BRCA1/2 mutational status on the efficacy of chemo-immunotherapy remains unclear, particularly in real-world settings. Since BRCA-mutated tumors exhibit homologous recombination deficiency (HRD) and high genomic instability, they may be more immunogenic and responsive to immune checkpoint inhibitors. This multicenter study investigated the association between BRCA1/2 mutations and pathologic complete response (pCR) in TNBC patients treated with pembrolizumab-based neoadjuvant chemotherapy (NACT). **Methods:** We retrospectively analyzed 184 patients with stage II–III TNBC treated between 2021 and 2024 across eleven Italian oncology centers. All received pembrolizumab combined with platinum- and taxane-based NACT followed by anthracyclines, according to the KEYNOTE-522 regimen. Germline BRCA1/2 status was determined by next-generation sequencing. The primary endpoint was pCR, defined as ypT0/is ypN0. Fisher’s exact test and logistic regression models were used to assess associations between clinical–pathological variables and pCR. **Results:** Among 184 patients, 25 (13.6%) harbored BRCA1 mutations, 12 (6.5%) BRCA2 mutations, and 147 (79.9%) were wild-type. pCR was achieved in 80.0% of BRCA1-mutated, 75.0% of BRCA2-mutated, and 61.1% of wild-type tumors. When pooled, BRCA1/2-mutated cases showed a higher likelihood of achieving pCR (78.4% vs. 61.1%; odds ratio [OR] = 2.17; 95% CI 1.01–4.97; *p* = 0.056). High tumor-infiltrating lymphocytes (≥30%) were also associated with increased pCR rates. The frequency of BRCA mutations (20.1%) was consistent with that reported in major TNBC series. No comparative analysis of toxicity or survival outcomes was performed due to the retrospective design and limited follow-up. **Conclusions:** In this multicenter real-world cohort, TNBC patients carrying BRCA1/2 mutations exhibited a trend toward higher pCR rates with pembrolizumab-based NACT compared with wild-type tumors. These findings suggest enhanced chemosensitivity and immune responsiveness in BRCA-deficient disease, warranting further validation in larger prospective studies with survival endpoints.

## 1. Introduction

Triple negative breast cancer (TNBC) continues to pose significant challenges for treatment, in the perioperative as well as the metastatic setting [1]. Despite progress in chemotherapy protocols such as the neo-adjuvant treatments, the implementation of dose-dense anthracyclines, and the incorporation of carboplatin, outcomes have still fallen short compared to other breast cancer types, particularly hormone receptor positive and HER2 positive disease [2,3,4]. This issue, combined with mounting evidence suggesting that TNBCs are more immunogenic, has resulted in several phase II and III clinical trials involving early TNBC patients, in which immunotherapeutics were associated with cytotoxic chemotherapy [5,6].

In recent years, pembrolizumab combined with chemotherapy has become a standard treatment for advanced PD-L1 positive TNBC and has also been included as part of neoadjuvant treatment for high-risk early TNBC, following the results of KEYNOTE-522 phase III clinical trial [7]. However, even with the practice-changing data brought about by KEYNOTE-522, crucial questions persist concerning the ideal candidate selection and the prediction of outstanding responders to chemoimmunotherapy [8,9]. The pathologic complete response (pCR) rate in TNBC patients receiving only chemotherapy is around 30–50%, something suggesting that a significant portion of patients could have achieved a pCR without the additional side effects of immunotherapy [10]. Thus, the identification of predictive biomarkers remains essential for directing patient selection, enhancing clinical results, and reducing the risk of chemoimmunotherapy adverse effects [11]. In fact, it is still unclear if specific patients undergoing neoadjuvant chemoimmunotherapy for TNBC may receive greater advantages from this treatment [12,13]. Since *BRCA*-mutated tumors display homologous recombination deficiency (HRD) and high genomic instability, they have been suggested to be more immunogenic and sensitive to immune checkpoint inhibitors. Based on these premises, we sought to evaluate the association between *BRCA1/2* mutations and pCR in a real-world cohort of TNBC patients treated with pembrolizumab-based neoadjuvant chemotherapy (NACT). The present analysis aimed to investigate the association between BRCA1/2 mutational status and pathologic complete response (pCR) in a real-world multicenter cohort of patients with triple-negative breast cancer (TNBC) treated with pembrolizumab-based neoadjuvant chemotherapy.

## 2. Materials and Methods

### 2.1. Study Population

This retrospective multicenter study included 184 consecutive patients with stage II–III TNBC treated between January 2021 and April 2024 across eleven Italian oncology centres. Eligible patients were aged ≥ 18 years and had histologically confirmed TNBC, defined as estrogen receptor (ER) and progesterone receptor (PR) expression < 1% and HER2 0–1+ or 2+ with negative in situ hybridization (ISH). Patients with de novo metastatic disease, pure metaplastic histology, prior systemic therapy for breast cancer, or missing pathological data were excluded.

All patients received pembrolizumab-based therapy following the KEYNOTE-522 regimen, which represents the current standard of care for high-risk early-stage TNBC after demonstrating a significant improvement in pCR (64.8% vs. 51.2%) and event-free survival in the phase III trial. The regimen consisted of pembrolizumab (200 mg every 3 weeks or 400 mg every 6 weeks) in combination with paclitaxel (80 mg/m^2^ weekly) and carboplatin (AUC 5 every 3 weeks or AUC 1.5 weekly) for 12 weeks, followed by doxorubicin (60 mg/m^2^) or epirubicin (90 mg/m^2^) with cyclophosphamide (600 mg/m^2^) every 3 weeks for four cycles. Surgery was performed 3–6 weeks after completion of neoadjuvant therapy, and adjuvant pembrolizumab was continued to complete one year of therapy whenever feasible, according to institutional practice and national guidelines.

Germline BRCA1 and BRCA2 mutational analyses were performed on peripheral blood DNA using next-generation sequencing (NGS) hereditary cancer panels validated for clinical use. Variant classification followed the ACMG/AMP 2015 criteria for the interpretation of sequence variants [14]. Only pathogenic or likely pathogenic variants were considered BRCA-mutated, whereas variants of uncertain significance (VUS) or benign variants were categorized as wild-type, consistent with current practice in clinical and translational BRCA-focused analyses. Testing was conducted in certified molecular pathology laboratories at each participating institution, ensuring standardized procedures for variant interpretation and reporting.

### 2.2. Study Endpoints

The primary endpoint was the rate of pCR according to BRCA1/2 mutational status. Pathologic complete response (pCR) was defined as ypT0/is ypN0, indicating the absence of residual invasive carcinoma in both breast and axillary lymph nodes, irrespective of residual in situ disease. Pathological evaluation was performed by institutional breast pathologists following international recommendations for post-neoadjuvant specimen assessment. Clinical and pathological data were retrospectively extracted from institutional electronic medical records and entered into a centralized anonymized database. Collected variables included demographic characteristics (age, menopausal status, body mass index), tumor features (clinical stage per AJCC, histologic grade, Ki-67 index, and tumor-infiltrating lymphocytes), and BRCA1/2 mutational status.

### 2.3. Statistical Analysis

Continuous variables were expressed as medians and interquartile ranges (IQR), and categorical variables as frequencies and percentages. Comparisons between groups (BRCA1-mutated, BRCA2-mutated, and wild-type) were performed using the Fisher’s exact test or χ^2^ test, as appropriate. Proportions were reported with 95% confidence intervals (CI) calculated by the Wilson method, which provides accurate interval estimation for binomial proportions in moderately sized samples. A two-sided *p*-value < 0.05 was considered statistically significant.

To improve interpretability and align with prior real-world series, BRCA1 and BRCA2 mutations were also analyzed together as a combined “BRCA-mutated” cohort, acknowledging the limited frequency of BRCA2 alterations in TNBC.

To explore clinical and pathological predictors of pCR, a univariate and multivariate logistic regression analysis was performed including the following covariates: age (continuous), clinical stage (II vs. III), ECOG performance status (0 vs. ≥1), body-mass index (BMI, continuous), tumor-infiltrating lymphocytes (TILs ≥ 30% vs. <30%) [14], and BRCA mutational status (mutated vs. wild-type). Variables with a *p* < 0.10 in univariate analysis were subsequently entered into the multivariate model. Odds ratios (OR) and 95% confidence intervals (CI) were estimated. A two-sided *p* < 0.05 was considered statistically significant. All analyses were performed using IBM SPSS Statistics v29.0 (IBM Corp., Armonk, NY, USA).

The study was approved by the Comitato Etico Territoriale Regione Puglia—Azienda Ospedaliero-Universitaria “Consorziale Policlinico” (study code 7889) and conducted in accordance with the Declaration of Helsinki. Informed consent was obtained when required by national and institutional regulations.

## 3. Results

A total of 184 patients with stage II–III TNBC treated with pembrolizumab-based neoadjuvant chemotherapy were included in this analysis. All patients were enrolled consecutively between January 2021 and April 2024 across eleven Italian oncology centers. The baseline clinical and pathological characteristics of the entire cohort are summarized in Table 1.

The median age at diagnosis was 48 years (range 42–55), and most of patients (n = 112, 60.9%) were pre-menopausal. The median body-mass index (BMI) was 24.6 kg/m^2^ (range 22.1–27.5). With respect to functional status, most patients had ECOG 0 (n = 152, 82.6%), while 32 patients (17.4%) presented with an ECOG performance status ≥ 1.

Regarding disease stage, 119 patients (64.7%) were classified as stage II, and 65 (35.3%) as stage III. Most tumors were high-grade (grade 3) lesions (n = 156, 84.8%). The median Ki-67 proliferation index was 70%, reflecting the typically high proliferative activity of TNBC.

The tumor microenvironment was characterized by variable immune infiltration. The median percentage of tumor-infiltrating lymphocytes (TILs) was 28% (range 15–45), and 82 patients (44.6%) had high TILs (≥30%), whereas 102 (55.4%) showed low infiltration (<30%). A total of 137 patients (74.5%) had experienced at least one pregnancy before diagnosis.

Comorbid conditions were documented in 86 patients (46.7%). Among these, thyroid disorders were the most frequent (21 patients, 11.4%), followed by autoimmune or inflammatory diseases (12 patients, 6.5%). Other comorbidities included hypertension (5, 2.7%), gastrointestinal disorders (4, 2.2%), hepatic disease (2, 1.1%), and single cases of cardiovascular, respiratory, renal, or psychiatric disorders (each 0.5%).

Germline BRCA testing was performed in all patients. Overall, 37 patients (20.1%) carried a pathogenic BRCA1/2 variant, including 25 BRCA1 (13.6%) and 12 BRCA2 (6.5%), while 147 patients (79.9%) were wild-type. The distribution of age, stage, and TIL levels was similar across BRCA-mutated and wild-type groups, with no significant baseline imbalances.

All patients received neoadjuvant treatment according to the KEYNOTE-522 regimen, consisting of pembrolizumab combined with platinum- and taxane-based chemotherapy followed by anthracycline and cyclophosphamide. Surgery was performed 3–6 weeks after the completion of chemotherapy, followed by adjuvant pembrolizumab administration in eligible patients according to institutional practice.

The overall pathologic complete response (pCR) rate in the study cohort was 63.6% (117 of 184 patients). Pathologic response data were available for all patients.

By subgroup, pCR was achieved in 20 of 25 BRCA1-mutated cases (80.0%; 95% CI 60.9–91.1), 9 of 12 BRCA2-mutated cases (75.0%; 95% CI 46.8–91.1), and 88 of 144 BRCA wild-type cases (61.1%; 95% CI 53.0–68.7). When BRCA1 and BRCA2 carriers were considered together, the combined BRCA1/2-mutated group achieved a pCR rate of 78.4% (95% CI 62.8–88.6), compared with 61.1% (95% CI 53.0–68.7) in the wild-type population. The difference approached statistical significance (*p* = 0.056, Fisher’s exact test).

Among the 67 patients (36.4%) who did not achieve pCR, 47 (25.5%) had residual invasive disease confined to the breast, while 20 (10.9%) presented with residual involvement of both breast and axillary lymph nodes. The pattern of residual disease did not differ substantially between BRCA-mutated and wild-type patients. The distribution of pCR outcomes according to BRCA1, BRCA2, and wild-type subgroups is summarized in Table 2 and depicted graphically in Figure 1, which shows the relative pCR proportions with 95% confidence intervals for each subgroup; pCR according to BRCA status were reported in Table 3.

In the univariate Cox model, BRCA mutations (OR 2.26; 95% CI, 1.00–5.12; *p* = 0.049) and high TILs (OR 1.98; 95% CI, 1.05–3.75; *p* = 0.034) were significantly associated with pCR; BRCA status was also associated with pCR in the multivariate model (Table 4) (OR 2.17; 95% CI, 1.01–4.97; *p* = 0.048).

## 4. Discussion

The introduction of pembrolizumab into the neoadjuvant setting has marked a turning point in the management of early-stage TNBC. Through this multicenter, retrospective analysis, we aimed to explore whether germline BRCA1/2 mutations could influence pCR to pembrolizumab-based NACT in everyday clinical practice. In our experience, BRCA-mutated tumors achieved numerically higher pCR rates compared with wild-type cases (78.4% vs. 61.1%), supporting a potential biological link between HRD and enhanced sensitivity to chemo-immunotherapy.

The overall pCR rate observed in our cohort (63.6%) closely mirrors the results of the KEYNOTE-522 trial (64.8%) and aligns with other real-world studies reporting rates between 45% and 65% [15,16]. Within this context, the apparent advantage among BRCA-mutated patients offers a clinically relevant observation. In our logistic regression model, BRCA mutations were associated with higher odds of achieving pCR (OR = 2.17; 95% CI 1.01–4.97; *p* = 0.056), reflecting a trend toward statistical significance. While this result does not meet conventional significance thresholds, we believe it remains biologically plausible given the well-known immunogenic profile of BRCA-deficient TNBC. These tumors typically display greater genomic instability, higher TIL levels, and increased neoantigen load factors that may contribute to heightened sensitivity to immune checkpoint blockade. In keeping with this hypothesis, high TILs (≥30%) were also associated with higher pCR rates in our cohort, in line with previous data underscoring immune activation as a predictive marker for ICI efficacy [17,18,19,20,21].

Several aspects of our analysis deserve careful interpretation. The relatively small number of BRCA2-mutated patients (n = 12) limits the power to discern distinct effects between BRCA1 and BRCA2. For this reason, we opted to analyze BRCA1/2 carriers as a combined group, though we acknowledge that this approach may have concealed potential subtype-specific differences. Moreover, as this was a retrospective study, granular data on dose intensity, treatment discontinuation, and immune-related adverse events were not systematically captured. In our clinical experience, some degree of treatment modulation is frequent in real-world settings, especially in younger patients balancing aggressive regimens with tolerability. These aspects, however, could not be formally evaluated here. Finally, survival outcomes such as EFS or OS were not assessed, as follow-up remains immature, and toxicity data were not reported. Future longitudinal analyses will help clarify whether the numerical advantage in pCR observed among BRCA-mutated patients translates into meaningful survival benefits.

In addition, we acknowledge some potentially relevant factors have not been included in some analysis, such as tumor size, nodal status, chemotherapy dose intensity, treatment delays, and PD-L1 expression. We are aware that the lack of some factors may have introduced some bias. Moreover, since toxicity, dose adjustment, and immune-related adverse events may have a strong influence on neoadjuvant therapy response, all these missing data may have biased pCR outcomes.

Despite these limitations, this study has important strengths. It represents one of the few real-world, multicenter efforts to address the relationship between BRCA status and immunotherapy response in TNBC. The inclusion of eleven Italian oncology centers reflects the collaborative nature of contemporary oncologic practice, where shared data can illuminate subtle biological and clinical patterns not easily discerned in single-institution experiences. The BRCA mutation frequency observed in our cohort (20.1%) closely parallels that of major TNBC series, reinforcing the representativeness and external validity of our population.

From a clinical standpoint, our findings highlight the importance of integrating germline BRCA testing into the neoadjuvant treatment pathway for TNBC. Identifying BRCA-mutated patients at diagnosis not only refines prognostic assessment but also facilitates a more informed discussion with patients regarding the potential for enhanced chemo-immunotherapy responsiveness. In our daily clinical practice, we often observe that BRCA-mutated TNBC exhibits a more rapid and profound response to systemic therapy. Nevertheless, the interplay between BRCA status, immune infiltration, and PD-L1 expression warrants deeper molecular investigation in prospective studies.

In conclusion, BRCA1/2-mutated TNBC patients treated with pembrolizumab-based NACT demonstrated a trend toward higher pCR rates compared with wild-type counterparts. These observations suggest that HRD may enhance both chemosensitivity and immune responsiveness.

## 5. Conclusions

In this multicenter real-world analysis, *BRCA1/2*-mutated TNBC showed numerically higher pCR rates with pembrolizumab-based NACT compared with wild-type tumors, suggesting enhanced chemosensitivity and immune responsiveness in *BRCA*-deficient disease. These findings underline the clinical value of *BRCA* testing to refine risk stratification and treatment decision-making in the neoadjuvant setting and further highlight the importance and potential clinical implications of BRCA testing in early TNBC, the role of BRCA status in neoadjuvant treatment decisions, and the need for prospective clinical trials that could validate these results.

## Figures and Tables

**Figure 1 jcm-14-08854-f001:**
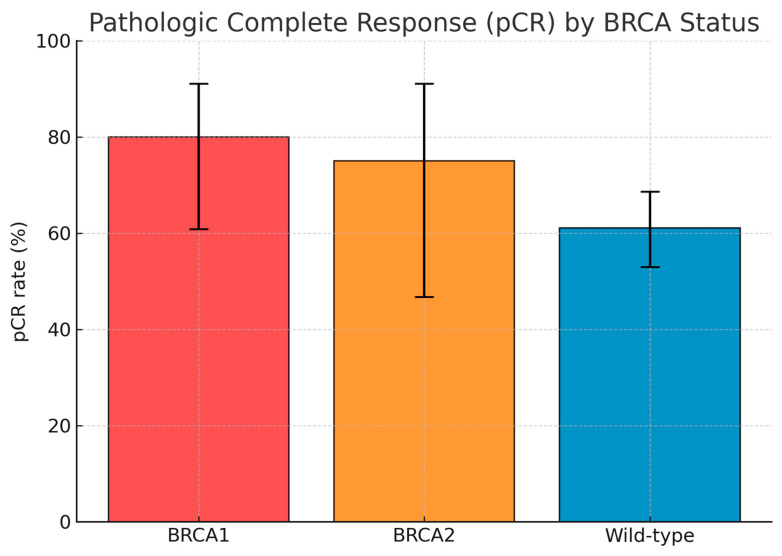
Pathologic complete response (pCR) by BRCA status. Bar chart illustrating the rate of pathologic complete response (pCR) according to BRCA mutational status in the study cohort (n = 184). pCR rates were higher in BRCA1- and BRCA2-mutated tumors (80.0% and 75.0%, respectively) compared with wild-type tumors (61.1%), with overlapping 95% confidence intervals (error bars). When pooled, BRCA1/2-mutated cases achieved a 78.4% pCR rate versus 61.1% in wild-type, showing a trend toward statistical significance (*p* = 0.056, Fisher’s exact test).

**Table 1 jcm-14-08854-t001:** Baseline characteristics of the study population (*n* = 184).

Variable	Value
Median age (years)	48 (IQR 42–55)
Median BMI (kg/m^2^)	24.6 (IQR 22.1–27.5)
ECOG 0	152 (82.6%)
ECOG ≥ 1	32 (17.4%)
BRCA-mutated	37 (20.1%)
BRCA1	25 (13.6%)
BRCA2	12 (6.5%)
BRCA wild-type	147 (79.9%)
Median TILs (%)	28 (IQR 15–45)
High TILs (≥30%)	82 (44.6%)
Low TILs (<30%)	102 (55.4%)
≥1 pregnancy	137 (74.5%)
Comorbidities—any	86 (46.7%)
Thyroid disease	21 (11.4%)
Autoimmune/inflammatory	12 (6.5%)
Hypertension	5 (2.7%)
Gastrointestinal	4 (2.2%)
Hepatic disease	2 (1.1%)
Cardiovascular disease	1 (0.5%)
Respiratory disease	1 (0.5%)
Renal disease	1 (0.5%)
Psychiatric disorder	1 (0.5%)

Abbreviations: BMI = body-mass index; TILs = tumor-infiltrating lymphocytes; IQR = interquartile range; ECOG = Eastern Cooperative Oncology Group performance status.

**Table 2 jcm-14-08854-t002:** Baseline characteristics according to BRCA status. Abbreviations: BMI = body-mass index; TILs = tumor-infiltrating lymphocytes; IQR = interquartile range; ECOG = Eastern Cooperative Oncology Group performance status.

Variable	BRCA-Mutated (*n* = 37)	BRCA Wild-Type (*n* = 147)	*p*-Value
Median age, years (IQR)	46 (41–52)	49 (43–56)	0.09
Stage III, n (%)	11 (29.7%)	54 (36.7%)	0.45
Median BMI (kg/m^2^)	24.3 (21.9–27.1)	24.7 (22.2–27.6)	0.68
ECOG ≥ 1, n (%)	5 (13.5%)	27 (18.4%)	0.49
High TILs (≥30%), n (%)	20 (54.1%)	62 (42.2%)	0.22
Median Ki-67 (%)	72 (60–80)	68 (50–75)	0.12
Comorbidities—any, n (%)	15 (40.5%)	71 (48.3%)	0.42
Pregnancy ≥ 1, n (%)	26 (70.3%)	111 (75.5%)	0.54

**Table 3 jcm-14-08854-t003:** Pathologic complete response (pCR) according to BRCA status. Abbreviations: Pathologic complete response (pCR) was defined as ypT0/is ypN0. BRCA1/2-mutated patients showed numerically higher pCR rates compared with wild-type cases, with a trend toward statistical significance (*p* = 0.056).

Group	*n*	pCR *n* (%)	95% Confidence Interval
BRCA1	25	20 (80.0%)	60.9–91.1
BRCA2	12	9 (75.0%)	46.8–91.1
Wild-type	144	88 (61.1%)	53.0–68.7
BRCA1/2 (combined)	37	29 (78.4%)	62.8–88.6

**Table 4 jcm-14-08854-t004:** Univariate and multivariate analysis for predictors of pathologic complete response (pCR). Abbreviations: Legend: OR = odds ratio; CI = confidence interval; TILs = tumor-infiltrating lymphocytes; BMI = body-mass index; ECOG = Eastern Cooperative Oncology Group performance status.

Variable	Univariate OR (95% CI)	*p*-Value	Multivariate OR (95% CI)	*p*-Value
BRCA mutation (mut vs. wt)	2.26 (1.00–5.12)	0.049	2.17 (1.01–4.97)	0.048
High TILs (≥30%)	1.98 (1.05–3.75)	0.034	1.75 (0.91–3.41)	0.09
Stage II vs. III	1.64 (0.87–3.09)	0.12	1.41 (0.73–2.86)	0.22
Age (continuous)	0.98 (0.94–1.02)	0.34	—	—
ECOG ≥ 1	0.79 (0.35–1.78)	0.57	—	—
BMI (continuous)	0.97 (0.89–1.04)	0.39	—	—

Variables with *p* ≥ 0.10 in univariate analysis were not included in the multivariate model (indicated by ‘—’).

## Data Availability

Due to the retrospective design and anonymization, the data are not shared publicly to protect patient privacy.

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
