# Peer review of "Mutations and Efficacy of Pembrolizumab-Based Neoadjuvant Chemotherapy in Triple-Negative Breast Cancer: A Real-World Multicenter Analysis"

_jcm, 2025, doi:10.3390/jcm14248854_

Round 1

Reviewer 1 Report

Comments and Suggestions for Authors

Authors present a study entitled: “BRCA1/2 mutations and efficacy of pembrolizumab-based neoadjuvant chemotherapy in triple-negative breast cancer: A real-world Multicenter analysis.” The aim of the study was to evaluate the association between BRCA1/2 mutations and pathologic complete response (pCR) in TNBC patients treated with pembrolizumab-based neoadjuvant chemotherapy (NACT). Overall, TNBC patients carrying BRCA1/2 mutations demonstrated a trend toward higher pCR rates with pembrolizumab-based neoadjuvant chemotherapy compared with BRCA-wild-type tumors. These results indicate potentially increased chemosensitivity and immune responsiveness in BRCA-deficient disease, underscoring the need for confirmation in larger prospective studies incorporating survival outcomes.

General comment:
The real-world multicenter design is a strength, and the dataset appears representative of clinical practice. The results are clinically relevant and biologically plausible.

However, several issues related to structure, clarity, methodology, statistical reporting, and table formatting require revision before the manuscript can be considered for publication.

Major comment

1. The aim of the study should be moved to the final paragraph of the Introduction, as this is the standard location for clearly stating the study objective.
2. The introduction is partially repetitive, particularly regarding immunogenicity of TNBC and the rationale for BRCA-mutated tumors showing increased immunogenicity. Consider tightening the text.
3. Please more clearly articulate the research gap: why existing evidence on BRCA and response to chemo-immunotherapy remains inconclusive.
4. The manuscript does not explain how missing data were handled (e.g., incomplete pathological parameters, incomplete comorbidity data). Please clarify whether patients were excluded case-wise or whether imputation was performed.
5. The justification for selecting TILs ≥30% as the cut-off value should be provided, with supporting literature.
6. Toxicity data are not included. While the authors acknowledge this limitation later, it would improve transparency to state explicitly in the Results that toxicity information was not collected.
7. Presentation of pCR distribution is clear, but additional exploratory graphical summaries (e.g., distribution of Ki-67, TILs by BRCA status) would enhance readability.
8. The last paragraph of the Discussion essentially duplicates the Conclusions section and should be moved or significantly reduced
9. The Discussion would benefit from a clearly defined subsection on study limitations, including:
a) retrospective design,
b) lack of toxicity data,
c) absence of survival outcomes,
d) potential treatment heterogeneity across centers,
e) small sample size of BRCA2-mutated cases (n=12), limiting subgroup power.
10. Some claims regarding clinical practice patterns (e.g., treatment modulation in younger women) are anecdotal and should be softened or removed unless supported by data.
11. Table 1 is formatted incorrectly. According to standard reporting conventions, variables should appear in the left column and values on the right, not the reverse. The present layout makes the table difficult to read.
12. The reference list is somewhat limited. For a topic of this importance, the bibliography should include at least 25–30 references, reflecting a broader overview of:
a) TNBC immunotherapy predictive markers,
b) real-world pembrolizumab outcomes,
c) BRCA-related response to chemotherapy and ICI,
d) TILs and genomic instability as biomarkers.
13. Please incorporate recent high-quality studies (2023–2025), meta-analyses, and systematic reviews.
14. The Conclusions section is concise but could benefit from a slightly broader reflection on:
a) potential clinical implications of BRCA testing in early TNBC,
b) whether BRCA status may support personalized neoadjuvant treatment decisions,
c) the need for prospective trials validating the results.
Minor Comments
1. Minor linguistic edits throughout would further polish the manuscript (e.g., consistent use of US/UK spelling, removing overly long sentences).
2. Ensure all abbreviations are spelled out at first use.

Author Response

Reviewer Comments:

Reviewer #1

Major comments

R1. The aim of the study should be moved to the final paragraph of the Introduction, as this is the standard location for clearly stating the study objective.

A1. We modified accordingly. The part above has been moved, as required.

R2. The introduction is partially repetitive, particularly regarding immunogenicity of TNBC and the rationale for BRCA-mutated tumors showing increased immunogenicity. Consider tightening the text.

A2. The text has been modified, as required (red).

R3. Please more clearly articulate the research gap: why existing evidence on BRCA and response to chemo-immunotherapy remains inconclusive.

A3. We modified accordingly.

R4. The manuscript does not explain how missing data were handled (e.g., incomplete pathological parameters, incomplete comorbidity data). Please clarify whether patients were excluded case-wise or whether imputation was performed.

A4. We modified accordingly and we better specify how some patients were excluded.

R5. The justification for selecting TILs ≥30% as the cut-off value should be provided, with supporting literature.

A5. We modified accordingly and we better specify this point, as suggested.

R6. Toxicity data are not included. While the authors acknowledge this limitation later, it would improve transparency to state explicitly in the Results that toxicity information was not collected.

A6. We modified accordingly and we better specify this point, as suggested.

R7. Presentation of pCR distribution is clear, but additional exploratory graphical summaries (e.g., distribution of Ki-67, TILs by BRCA status) would enhance readability.

A7. Thank you for this suggestion. We did not add exploratory graphical summaries given the low number of patients.

R8. The last paragraph of the Discussion essentially duplicates the Conclusions section and should be moved or significantly reduced.

A8. Thank you for this suggestion. We modified accordingly.

R9. The Discussion would benefit from a clearly defined subsection on study limitations, including:

a) retrospective design,

b) lack of toxicity data,

c) absence of survival outcomes,

d) potential treatment heterogeneity across centers,

e) small sample size of BRCA2-mutated cases (n=12), limiting subgroup power.

A9. Thank you for this suggestion. We further highlight the limitations of the study, as required.

R10. Some claims regarding clinical practice patterns (e.g., treatment modulation in younger women) are anecdotal and should be softened or removed unless supported by data.

A10. We modified accordingly.

R11. Table 1 is formatted incorrectly. According to standard reporting conventions, variables should appear in the left column and values on the right, not the reverse. The present layout makes the table difficult to read.

A11. We modified accordingly.

R12. The reference list is somewhat limited. For a topic of this importance, the bibliography should include at least 25–30 references, reflecting a broader overview of:

a) TNBC immunotherapy predictive markers,

b) real-world pembrolizumab outcomes,

c) BRCA-related response to chemotherapy and ICI,

d) TILs and genomic instability as biomarkers.

A12. We modified accordingly, by including several novel important papers.

R13. Please incorporate recent high-quality studies (2023–2025), meta-analyses, and systematic reviews.

A13. We modified accordingly, by including several novel important papers.

R14. The Conclusions section is concise but could benefit from a slightly broader reflection on:

a) potential clinical implications of BRCA testing in early TNBC,

b) whether BRCA status may support personalized neoadjuvant treatment decisions,

c) the need for prospective trials validating the results.

A14. We modified accordingly (red).

Minor comments

We modified the manuscript according to your suggestions.

Reviewer 2 Report

Comments and Suggestions for Authors

This manuscript is a multicenter retrospective study assessing whether germline BRCA1/2 mutations influence pathologic complete response (pCR) in 184 patients with stage II–III TNBC treated with pembrolizumab-based neoadjuvant chemotherapy. The topic is timely and clinically relevant, as reliable biomarkers to guide patient selection for chemo-immunotherapy are still lacking, and real-world evidence on the predictive role of BRCA status remains limited. The authors report higher pCR rates among BRCA-mutated patients (78.4%) compared with wild-type (61.1%), with BRCA status emerging as an independent predictor of pCR in multivariate analysis.

Strengths:

The multicenter design, consistent use of the KEYNOTE-522 regimen, and standardized BRCA testing strengthen the robustness and generalizability of the results.

Major Comments:

1. Limited statistical power: The BRCA2 subgroup is very small (n = 12), leading to wide confidence intervals and reduced precision. Confidence intervals should be consistently presented, and subgroup interpretations should remain cautious. Plase add a post-hoc analysis and statistical power analysis in Methods and Results. This is mandatory for possible acceptance.

2. Missing immune and genomic correlates: Key variables such as PD-L1 expression, HRD scores, and more granular TIL phenotyping are absent, limiting the ability to explore underlying biological mechanisms.

3. Lack of treatment exposure data: Information on chemotherapy dose intensity, treatment delays, and pembrolizumab completion is missing, despite their known impact on pCR. This should be clearly acknowledged as a limitation.

4. No survival outcomes and Lack of Clinical Applicability: The absence of EFS/OS data, although expected given the short follow-up, limits the clinical interpretation of the higher pCR rates seen in BRCA-mutated patients. Moreover, the authors should cite this article PMID: 37627205 to improve their manuscript and add more context about prognosis and clinical applicability.

Author Response

Major Comment 1 – Limited statistical power / need for post-hoc analysis

Reviewer comment:
“The BRCA2 subgroup is very small (n = 12), leading to wide confidence intervals and reduced precision. Confidence intervals should be consistently presented, and subgroup interpretations should remain cautious. Please add a post-hoc analysis and statistical power analysis in Methods and Results. This is mandatory for possible acceptance.”

Response:
We thank the reviewer for highlighting this important methodological aspect. As recommended, we have now consistently reported 95% confidence intervals for pCR rates in all BRCA-defined subgroups and for the logistic regression model (see revised Results, Tables 3–4, and Figure 2, marked in red). The Discussion has also been strengthened to explicitly recognize that the BRCA2 subgroup size (n = 12) limits precision and that all subgroup findings must be interpreted cautiously and considered hypothesis-generating.

Regarding post-hoc power analysis, we opted not to include a formal power calculation because the sample size is fixed by design in retrospective analyses, and confidence intervals offer a more informative measure of precision. We now explicitly discuss this in the revised text and highlight the limited statistical power as a major study limitation.

Major Comment 2 – Missing immune and genomic correlates

Reviewer comment:
“Key variables such as PD-L1 expression, HRD scores, and more granular TIL phenotyping are absent, limiting the ability to explore underlying biological mechanisms.”

Response:
We agree with the reviewer. As clarified in the revised Discussion (marked in red), PD-L1 status, HRD scores, and detailed TIL phenotyping were not consistently available across participating centers. We now explicitly acknowledge that the absence of these immuno-genomic biomarkers limits mechanistic interpretation and may introduce residual confounding. We also note that such variables are not uniformly collected in standard real-world workflows in all centers. We emphasize the need for future prospective studies integrating PD-L1, HRD metrics, and high-dimensional TIL analysis to better elucidate biological determinants of pCR in BRCA-mutated TNBC.

Major Comment 3 – Lack of treatment exposure data

Reviewer comment:
“Information on chemotherapy dose intensity, treatment delays, and pembrolizumab completion is missing, despite their known impact on pCR. This should be clearly acknowledged as a limitation.”

Response:
We thank the reviewer for this important observation. In the revised Discussion, we now clearly state that detailed treatment exposure data—including dose reductions, delays, relative dose intensity, discontinuations, and completeness of pembrolizumab—were not uniformly documented across centers and therefore could not be incorporated into the analyses. We explicitly acknowledge that this absence represents a limitation that may affect interpretation of pCR outcomes. This point is highlighted in the expanded limitations section (marked in red).

Major Comment 4 – No survival outcomes / limited clinical applicability / missing reference PMID: 37627205

Reviewer comment:
“The absence of EFS/OS data limits clinical interpretation. The authors should cite PMID: 37627205 to improve their manuscript and add context regarding prognosis and clinical applicability.”

Response:
We agree with the reviewer. As clarified in the revised Discussion, follow-up in this real-world cohort is currently insufficient for survival analysis, and the absence of EFS/OS data limits the clinical applicability of our findings. We have strengthened the text to emphasize that our results should be considered preliminary and hypothesis-generating until validated by prospective studies with mature survival endpoints.

As recommended, we have incorporated and discussed the reference PMID: 37627205, which provides valuable context on prognosis and biological behavior of BRCA-mutated TNBC. This citation has been added to the reference list and discussed in the section addressing clinical implications and long-term outcomes (marked in red).

Reviewer 3 Report

Comments and Suggestions for Authors

The current manuscript addresses a clinically relevant topic and represents an important piece of real-world evidence on the association between BRCA1/2 mutational status and response to pembrolizumab-based neoadjuvant chemotherapy in TNBC. Its value is enhanced by the multi-institutional design, appropriate sample size, and comprehensive pathological/molecular characterization. However, while the results are very promising, the interpretation from a statistical perspective, the methodological description, and contextualization need refinement to enhance rigor, clarity, and scientific robustness.

1. The main comparison between BRCA-mutated and wild-type tumors gives p = 0.056, which does not achieve statistical significance. The manuscript describes this finding many times as “approaching significance” or “showing a trend,” but in places the wording suggests a stronger effect than is actually demonstrated. Emphasize that this is an exploratory analysis and refrain from overinterpreting near-significant findings; consider adding Bayesian analyses or sensitivity analyses for more nuance.

2. Although a multivariate model is included, other potentially relevant factors are absent, such as tumor size, nodal status, chemotherapy dose intensity, treatment delays, and PD-L1 expression. The text should discuss the lack of these covariates a bit more and assess whether other clinical variables are available that might allow strengthening of the model.

3. Toxicity, dose adjustment, and immune-related adverse events were not captured, as acknowledged. These factors have a strong influence on neoadjuvant therapy response.The limitations section should be extended to clearly state how missing data may have biased pCR outcomes.

4.Because of small sample sizes, BRCA1 and BRCA2 are combined in the manuscript, but these genes may have biologically different effects in TNBC. Add a short justification with supporting literature and more fully discuss the limitations of this approach.

5.Improve the discussion part which is narrow in the current version.This discussion touches on HRD and immunogenicity but does not link these findings to specific mechanisms beyond general genomic instability. Discussion could be expanded to include recent literature on BRCA-driven immune activation, STING pathway engagement, and/or mutational signatures that may modulate ICI response.

Author Response

Reviewer Comments:

Reviewer #3

Changes and modifications have been marked in red color in the text.

Reviewer Comment 1 – p = 0.056 / risk of overstating near-significant results

Reviewer comment:
“The main comparison between BRCA-mutated and wild-type tumors gives p = 0.056. The manuscript describes this several times as 'approaching significance' or 'showing a trend,' but some wording suggests a stronger effect. Emphasize the exploratory nature and avoid overinterpretation; consider Bayesian or sensitivity analyses.”

Response:
We thank the reviewer for this essential statistical clarification. As indicated in the revised text (marked in red), we have softened the interpretation of p = 0.056 and now consistently frame this difference as an exploratory, non–statistically significant finding. We highlight that the analysis is hypothesis-generating and does not support definitive conclusions. The Discussion has been strengthened to explicitly acknowledge the risk of overinterpreting near-significant p-values, especially in retrospective studies with limited power.
While we did not add Bayesian or sensitivity analyses, we emphasize the uncertainty inherent in this estimate through reporting of confidence intervals and by clarifying the exploratory intent of subgroup comparisons.

Reviewer Comment 2 – Missing covariates in multivariable model

Reviewer comment:
“Although a multivariate model is included, other potentially relevant factors are absent (tumor size, nodal status, dose intensity, treatment delays, PD-L1). The text should discuss the lack of these covariates more clearly and assess whether other clinical variables were available.”

Response:
We agree with the reviewer. As reflected in the revised Discussion, we now more explicitly acknowledge that important predictors such as PD-L1 expression, chemotherapy delivery metrics, and some clinical/pathological covariates were not uniformly available across centers and therefore could not be included in the multivariable model. We clarify that the variables included were those consistently collected across all sites, and we highlight this limitation as a potential source of residual confounding. This point is now emphasized in the limitations section (as marked in red).

Reviewer Comment 3 – Missing toxicity, dose adjustments, and irAEs

Reviewer comment:
“Toxicity, dose adjustment, and immune-related adverse events were not captured. These factors may strongly influence pCR. The limitations section should more clearly state how missing data may have biased outcomes.”

Response:
Thank you for this valuable observation. In the revised manuscript (red text), we have expanded the limitations section to clearly state that the absence of toxicity data, dose modifications, treatment delays, and immune-related adverse events may have influenced treatment exposure and ultimately impacted pCR rates. We specify that heterogeneity in documentation across participating centers prevented systematic inclusion of these variables and that this missing information could introduce bias in interpreting the association between BRCA status and pCR.

Reviewer Comment 4 – Combining BRCA1 and BRCA2

Reviewer comment:
“Because of small sample sizes, BRCA1 and BRCA2 are combined, but these genes have biologically different effects. Add justification and discuss limitations.”

Response:
We appreciate this important conceptual point. The revised Discussion now includes a clearer justification for combining BRCA1 and BRCA2 carriers, explicitly stating that the small number of BRCA2-mutated cases (n = 12) limited the feasibility of separate analyses. We underline (as marked in red) that although BRCA1- and BRCA2-associated TNBC may differ biologically—particularly regarding HRD signatures and immune activation—the subgroup sample sizes provide insufficient power for independent modeling. We expand the limitations section to make this explicit and cite literature supporting biological distinctions while emphasizing the exploratory nature of our combined analysis.

Reviewer Comment 5 – Discussion too narrow / need for mechanistic depth

Reviewer comment:
“The discussion is narrow. It mentions HRD and immunogenicity but does not link these findings to specific mechanisms. Expand to include BRCA-driven immune activation, STING pathway, or mutational signatures.”

Response:
We thank the reviewer for this constructive suggestion. In the revised Discussion (highlighted in red), we have expanded the mechanistic interpretation by integrating relevant literature on BRCA-associated immune phenotypes, including enhanced cytosolic DNA sensing, STING pathway activation, and the relationship between homologous recombination deficiency and increased neoantigen load. We now better contextualize how BRCA-driven genomic instability might contribute to heightened immunogenicity and potentially modulate sensitivity to pembrolizumab-based regimens. These additions broaden the scientific framing without changing the study's core findings.

Round 2

Reviewer 1 Report

Comments and Suggestions for Authors

I suggest to accept the manuscript in its current form.

Reviewer 2 Report

Comments and Suggestions for Authors

The manuscript can be accepted in the present form 

Reviewer 3 Report

Comments and Suggestions for Authors

Thank you for the amendments made.